# In vivo mitochondrial base editing via adeno-associated viral delivery to mouse post-mitotic tissue

Pedro Silva-Pinheiro [1], Pavel A. Nash[1], Lindsey Van Haute [1], Christian D. Mutti [1], Keira Turner [1] & Michal Minczuk [1]✉

Mitochondria host key metabolic processes vital for cellular energy provision and are central to cell fate decisions. They are subjected to unique genetic control by both nuclear DNA and their own multi-copy genome - mitochondrial DNA (mtDNA). Mutations in mtDNA often lead to clinically heterogeneous, maternally inherited diseases that display different organ-specific presentation at any stage of life. For a long time, genetic manipulation of mammalian mtDNA has posed a major challenge, impeding our ability to understand the basic mitochondrial biology and mechanisms underpinning mitochondrial disease. However, an important new tool for mtDNA mutagenesis has emerged recently, namely double-stranded DNA deaminase (DddA)-derived cytosine base editor (DdCBE). Here, we test this emerging tool for in vivo use, by delivering DdCBEs into mouse heart using adeno-associated virus (AAV) vectors and show that it can install desired mtDNA edits in adult and neonatal mice. This work provides proof-of-concept for use of DdCBEs to mutagenize mtDNA in vivo in post-mitotic tissues and provides crucial insights into potential translation to human somatic gene correction therapies to treat primary mitochondrial disease phenotypes.

---

[1] MRC Mitochondrial Biology Unit, University of Cambridge, Cambridge CB2 0XY, UK. ✉email: michal.minczuk@mrc-mbu.cam.ac.uk

Mitochondria play a central role in energy provision to the cell and in several key metabolic pathways, such as thermogenesis, calcium handling, iron-sulfur cluster biogenesis, and apoptosis[1,2]. Mitochondria produce energy in the form of ATP, which is synthesized in the process of oxidative phosphorylation (OXPHOS), involving sequential redox reactions coupled with proton pumping performed by mitochondrial membrane-embedded respiratory chain complexes (I–IV) and ATP synthase (complex V). In mammals, the mitochondrial proteome comprises ~1200 proteins[3], with most of them being nuclear DNA (nDNA)-encoded. However, 13 essential OXPHOS polypeptides, and 22 tRNAs, and 2 rRNAs required for their translation reside inside the mitochondrial matrix, encoded by the mitochondrial DNA (mtDNA)—a 16.5 kb, maternally inherited multicopy circular genome.

Mitochondrial diseases are genetic disorders, caused by mutations, either in nDNA or mtDNA, that lead to mitochondrial energy production impairment and perturbations in other aspects of cellular homeostasis. With a prevalence of ~23 in 100,000, mitochondrial disorders are among the most common inherited diseases, and are often associated with severe disability and shortened lifespan[4]. There are currently no effective treatments for these disorders and clinical management focuses on treating complications[5]. Mutations in mtDNA and mitochondrial dysfunction have also been implicated in many common diseases with high societal impact and ageing[6,7]. Mammalian cells can contain 100–1000 s copies of mtDNA[8]. Pathogenic variants in mtDNA can either be present in all copies (homoplasmy) or only in a portion of genomes (heteroplasmy), with mutant load varying across cells, tissues, and organs[9]. In heteroplasmic cells, the mutant load required for clinical expression must exceed a threshold, which is highly variable and is dependent on the mtDNA variant, affected tissues/organ(s), and genetic/environmental contexts, but is usually more than 60%[10].

Despite the ongoing genome-engineering revolution enabled by the CRISPR/Cas-based systems, mammalian mtDNA has been resistant to genetic modifications, in a large part owing to ineffective nucleic acid import into the mitochondria[11]. The inability to edit mtDNA sequences in mammalian mitochondria within cells has hampered the research of normal mtDNA processes, the development of in vivo models and therapies for mtDNA diseases. For many years the approaches towards manipulation of mtDNA in mammals have been mainly limited to mitochondrially targeted restriction enzymes[12–14] and programmable nucleases[15–21]. These nucleases have been used to eliminate undesired mtDNA molecules from heteroplasmic populations, to move the mutant mtDNA heteroplasmy below the pathogenicity threshold[22]. Following extensive trials in vitro, the mitochondrial nuclease-based approaches have reached in vivo proof-of-concept. Delivered by adeno-associated virus (AAV) in heteroplasmic mouse models, mtRE, mtZFN, mitoTALEN, or mitoARCUS demonstrated specific elimination of the mutant mtDNA in the target tissues[23,24], which in some models and approaches was accompanied by the molecular and physiological rescue of disease phenotypes[25–27].

While programmable nucleases have proven useful in changing the existing heteroplasmy, they are unable to introduce novel mtDNA variants. However, recently a novel tool has emerged: DddA-derived cytosine base editor (DdCBE), which catalyzes site-specific C:G to T:A conversions in mtDNA with good target specificity in human cultured cells[28]. DdCBE is based upon a modified bacterial toxin DddA$_{tox}$ (separated, non-toxic halves fused to TALE proteins) which is targeted to the mitochondrial matrix to catalyze the deamination of cytidines within dsDNA at sequence determined by TALE design[28]. Current DdCBEs deaminate cytidines (in the TC:GA sequence context) to uracil

leading to a TC:GA > TT:AA mutations upon subsequent replication[28]. An initial proof-of-concept of successful installment of mtDNA edits by delivering DdCBEs mRNA into embryos in the mouse[29] and zebrafish[30] has also recently been provided.

In this study, we provide proof-of-concept for the use of DdCBEs in vivo in somatic tissue. We used the mouse heart as a surrogate of a post-mitotic tissue and showed that DdCBE delivered using AAV can install the desired mtDNA mutations in adult and neonatal mice. To the best of our knowledge, such a result has not been reported in the literature thus far. This work demonstrates that the DdCBE platform could be used for tissue-specific mtDNA mutagenesis in vivo and potentially for future therapies based upon somatic mitochondrial gene correction to treat mtDNA-linked mitochondrial diseases.

## Results

### Design of DdCBE and mtDNA editing in mouse cultured cells.
With the intention of testing the emerging mtDNA editing DdCBE technology, we set off to induce de novo mutations in mouse mitochondrial complex I in cultured cells and in somatic tissues upon AAV delivery. We aimed at editing the GGA glycine 40 codon in mouse *MT-Nd3* (mtDNA positions: m.9576 G and m.9577 G) by targeting the complementary cytosine residues with DdCBE (Fig. 1a, $C_{12}$ $C_{13}$). To enable this, we designed four DdCBE pairs, containing TALE domains binding the mtDNA light (L) and heavy (H) strands (mtDNA positions m.9549–m.9564 and m.9584–m.9599, respectively) and different combinations of DddA$_{tox}$ splits (G1333 or G1397), targeting a 19 bp-long sequence in the mouse *MT-Nd3* gene (mtDNA positions: m.9565–m.9583) (Fig. 1a-b) and named them DdCBE-Nd3-9577-1 to 4. The previous study has shown that for the sites containing two consecutive cytosines (preceded by a thymine), both of these cytosines can be edited by DdCBEs, with the following potential consequences TCC:GGA > TTC:GAA, TCC:GGA > TCT:AGA, TCC:GGA > TTT:AAA (edited C underlined)[28]. In line with this, we predicted three possible outcomes of m.9576 G and m.9577 G editing (Fig. 1b): [i] deamination of both complementary cytosines (Fig. 1b, $C_{12}$ and $C_{13}$), would result in glycine to lysine mutation (G40K), [ii] deamination of $C_{13}$ would lead to a glycine to glutamic acid mutation (G40E), whereas [iii] exclusive editing of $C_{12}$ would lead to a premature AGA stop codon (G40*), according to the mitochondrial genetic code (Fig. 1b). The three predicted mutations are located in the conserved ND3 loop, involved in active/deactive state transition of complex I (Fig. 1c-d)[31,32].

Next, we transiently delivered DdCBE-Nd3-9577 pairs into mouse cultured NIH/3T3 cells and selected the transfectants using fluorescence-activated cell sorting (FACS) at 24 h post-transfection and allowed for a 6 day-long recovery (Fig. 2a). After 7 days, we detected efficient editing of target cytosines by Sanger sequencing for the DdCBE-Nd3-9577 pairs 1, 2 and 3 (Fig. 2b). Although the targeted 19 bp sequence region between the TALE binding sites contained five cytosine residues with the correct thymine-cytosine (TC) consensus (Fig. 1a, $C_6$, $C_{12}$, $C_{13}$, $C_{17}$, and $C_{18}$), positioning of the DddA$_{tox}$ deaminase domain allowed for editing of $C_{12}$, $C_{13}$ leading to the expected changes in the *MT-Nd3* GGA glycine 40 codon (Fig. 2b). Next, we performed next-generation sequencing (NGS) analysis of the DdCBE-Nd3-9577-mediated mutagenesis to quantify the editing in each target cytosine and measure the proportion of editing resulting in the G40K, G40E, G40* mutations. This analysis revealed up to ~43% editing of $C_{12}$ and $C_{13}$ for DdCBE-Nd3-9577 pair 1, ~20–35% editing for pairs 2 and 3, and confirmed negligible editing activity of pair 4 (Fig. 2c). Furthermore, most of the NGS reads for pair 1 contained editing of both $C_{12}$ and $C_{13}$ translating to G40K mutation (~92.5%), whereas DNA reads corresponding to the

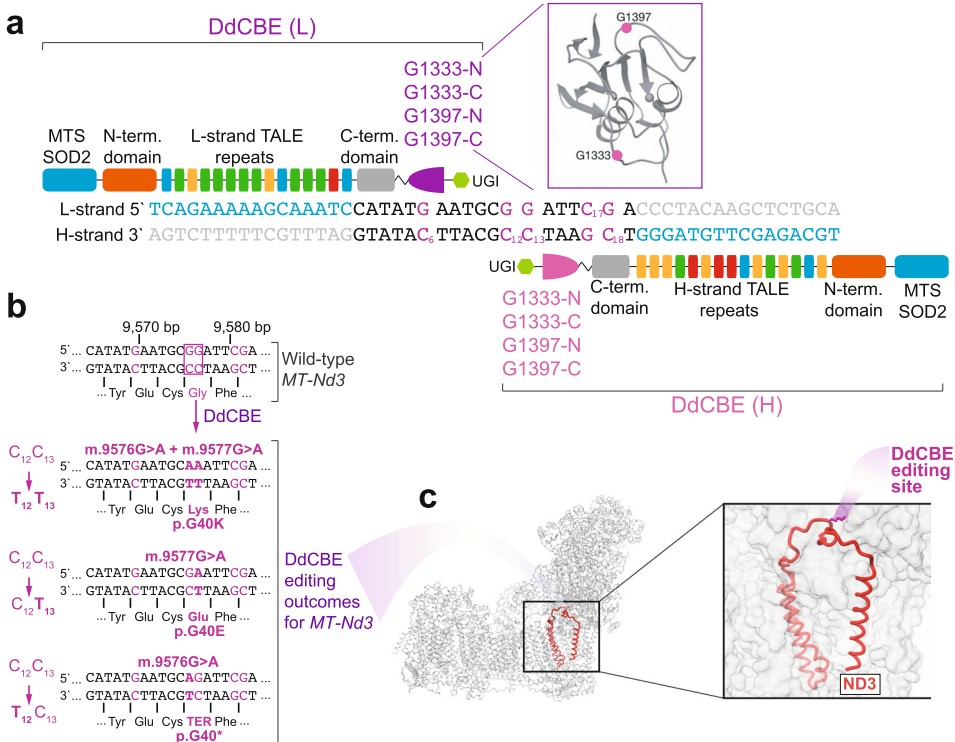

**Fig. 1 Design of DdCBEs and mutagenesis site. a** The architecture of DdCBE monomers targeting m.9576 G (C12) and m.9577 G (C13). The DNA specificity is provided by TALE domains. In each experiment, different DddA_tox splits are used (G1397 or G1333, purple) to achieve editing of "TC" sites. MTS SOD2 mitochondrial targeting sequence from superoxide dismutase 2, UGI uracil glycosylase inhibitor, L-strand or (L) light mtDNA strand, H-strand or (H) heavy mtDNA strand. **b** The details and possible outcomes of m.9576 G (C12) and m.9577 G (C13) editing. The purple box indicates the desired editing sites; other potential editing sites are indicated in purple. **c** The structural model of mouse complex I with indicated MT-ND3 subunit (red). The inset shows the location of MT-ND3 p.G40K mutation on the evolutionary conserved MT-ND3 loop.

G40E and G40* changes constituted only ~4.5% and 3%, respectively (Fig. 2d). A similar editing pattern, but with a higher proportion of reads corresponding to G40E, was observed for pair 3 (G40K: ~83%, G40E: 14% G40*: 3%) (Fig. 2d). However, the mutagenesis pattern detected for pair 2 was skewed towards G40E, as compared to pairs 1 and 3, with reads corresponding to G40E accounting for 45.5% of C12 and C13-edited reads (G40K: ~53% and G40*: 1.5%) (Fig. 2d). The NGS analysis revealed a low level (below 3%) of C6 (mtDNA position m.9570 G) for pairs 1–3, which is predicted to install the E38K change (Fig. 2c). To confirm that the observed mtDNA editing is indeed a result of the catalytic activity of DddA_tox, we used the DdCBE-Nd3-9577 pairs harboring a catalytically inactive DddA_tox (E1347A) in a control transient transfection experiment. The NGS analysis showed that none of the catalytically inactive DdCBEs exerted detectable deamination activity that would lead to mtDNA editing, with the mutagenesis frequency being at the level of wild-type cells (Fig. 2c). Based on these in vitro results, we concluded that the DdCBE-Nd3-9577-1 pair is the most suitable for efficient installation of G40K and decided to proceed with in vivo experiments using this set.

**AAV-based in vivo DdCBE editing of mtDNA in adult mice.** To provide a proof-of-concept for in vivo mtDNA gene editing of somatic cells, we encapsidated the catalytically active and inactive versions of the DdCBE-Nd3-9577-1 monomers into the cardiotropic AAV9.45 serotype and administered them systemically via tail-vein injection at $1 \times 10^{12}$ viral genomes (vg) per monomer per 8-week-old adult mouse (Fig. 3a). At 3- and 24-weeks post-injection, we confirmed successful DdCBE DNA delivery to the cardiac tissue by quantitative PCR (Supplementary Fig. 1a-b) and detected

its expression in total mouse heart tissue by western blotting (Supplementary Fig. 1c) and immunohistochemistry (Supplementary Fig. 1d-e). At 3 weeks after DdCBE-Nd3-9577-1 AAV injections, we detected low-level editing (1–2%) of the target C12 and C13 bases (corresponding to m.9576G > A and m.9577G > A) by NGS, but not by Sanger sequencing (Fig. 3b-c). The NGS analysis revealed C12 and C13 mutagenesis in the hearts of animals injected with catalytically active DdCBE-Nd3-9577-1, but not in those injected with a vehicle or catalytically inactive DdCBE, with the C12 and C13 editing pattern resembling the one observed in in vitro experiments (Fig. 3d). Notwithstanding, at 24-weeks post-injection robust editing (10–20%) of C12 and C13 was observed in the cardiac tissue of mice injected with catalytically active DdCBE-Nd3-9577-1 by Sanger sequencing and NGS, with no detectable changes being detected for vehicle and catalytically inactive controls (Fig. 3e-f). Further analysis confirmed that in the majority (94%) of the NGS reads both C12 and C13 were edited, translating into the G40K MT-Nd3 mutation (Fig. 3g). We did not detect any adverse effect on mtDNA copy number in DdCBE AAV-transduced cardiac tissue as compared to vehicle-injected controls (Supplementary Fig. 1f-g). Taken together, these results established that DdCBE editing can be applied to install mtDNA mutations in vivo in post-mitotic tissues.

**AAV-based in vivo DdCBE editing of mtDNA in neonates.** Having established that de novo mutations can be installed in vivo in the post-mitotic tissue of adult mice following AAV-assisted DdCBE transduction, we set off to investigate whether transgene delivery to younger subjects could enhance the editing. To this end, we injected neonatal subjects (first 24 h of life) with DdCBE-Nd3-9577-1 AAV9.45 and its catalytically inactive version via temporal vein injection at $1 \times 10^{12}$ vg per monomer per animal (Fig. 4a).

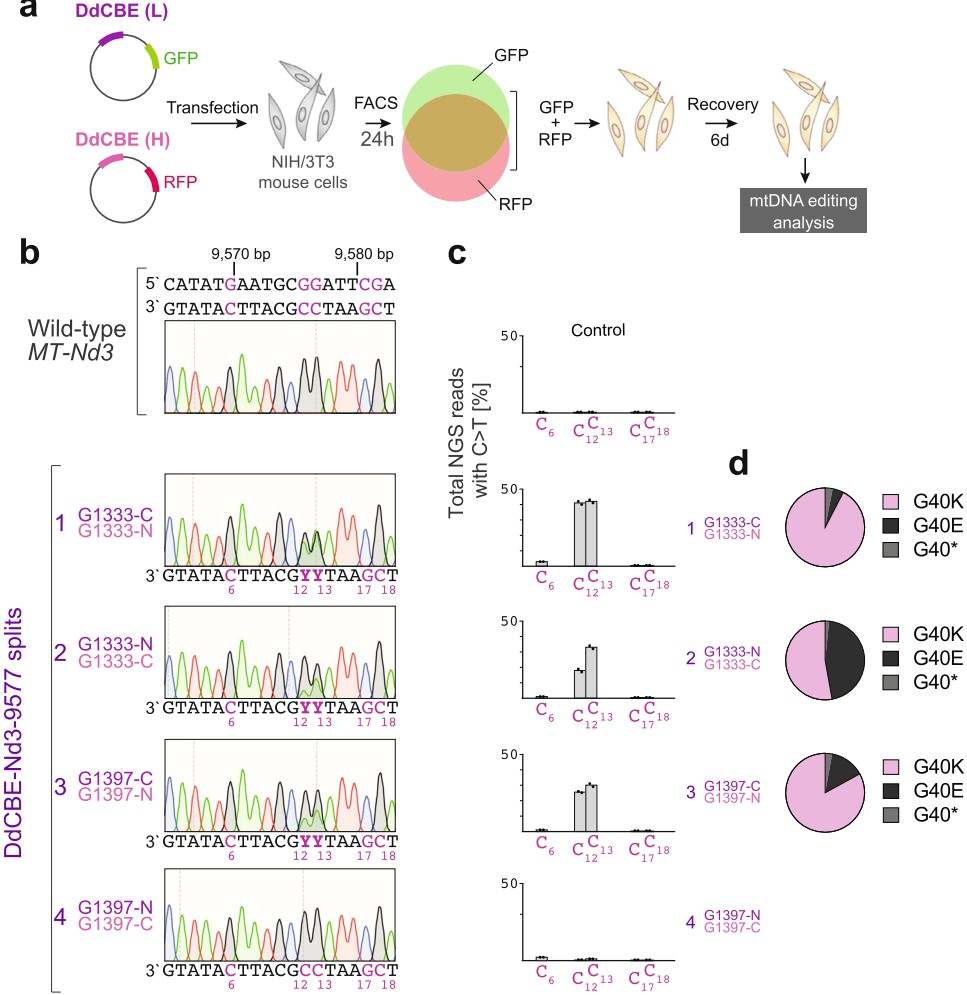

**Fig. 2 Mitochondrial DNA editing in cultured mouse cells. a** Schematic of the general workflow for in vitro experiments that involve transient transfection of cultured mouse NIH/3T3 cells with plasmids co-expressing DdCBE monomers and fluorescent marker proteins, FACS-based selection of cells expressing both monomers and evaluation of mtDNA from DdCBE-treated cells. **b** Editing of mouse *MT-Nd3* with different DdCBE splits in cells 7 days post-transfection, analyzed by Sanger sequencing. Potential editing sites are indicated in purple. The C:G > T:A deamination leads to m.9576 G > A ($C_{12} > T_{12}$) and m.9577 G > A ($C_{13} > T_{13}$) mutations, which translate to p.G40K change in MT-ND3. **c** The NGS analysis of the editing region in cells treated with different DdCBE splits. Bars represent the mean (*n* = 2). Source data are provided as a Source Data file. The mutagenesis frequency for the catalytically inactive versions is provided in the Source Data file. **d** The distribution of NGS reads containing m.9576 G ($C_{12}$) or m.9577 G ($C_{13}$) edits. The G40K reads contain both m.9576 G > A ($C_{12} > T_{12}$) and m.9577 G > A ($C_{13} > T_{13}$) mutations, G40E reads contain only the m.9577 G > A ($C_{13} > T_{13}$) mutation, while G40* reads contain only the m.9576 G > A ($C_{12} > T_{12}$) mutation. Source data are provided as a Source Data file.

Upon sacrificing the animals at 3-weeks post-injection, we confirmed successful DdCBE-Nd3-9577-1 AAV delivery (Supplementary Fig. 2a) and robust expression in the total mouse heart (Supplementary Fig. 2b-c). At this time-point we observed high efficiency of editing of mtDNA in mouse heart, with Sanger sequencing and NGS revealing 20–30% of C > T (G > A) changes of $C_{12}$ and $C_{13}$ in the DdCBE-Nd3-9577-1-targeted spacing region (Fig. 4b-c). No editing of these bases was observed in the control pups injected with a vehicle and the catalytically inactive DdCBE-Nd3-9577-1 AAVs (Fig. 4c). The vast majority of NGS reads (95%) contained simultaneous edits of $C_{12}$ and $C_{13}$ (corresponding to m.9576 G > A and m.9577 G > A), translating into the G40K *MT-ND3* mutation (Fig. 4d). We did not observe any significant changes in mtDNA copy number in the AAV-treated mice as compared to the vehicle-injected controls (Supplementary Fig. 2d). Taken together, these data not only further confirm that DdCBE-mediated mtDNA editing is possible in post-mitotic tissues upon AAV delivery, but also show that treatment of the younger subjects is beneficial for the efficacy of mtDNA modification.

**Off-target editing by DdCBE in adult and neonatal mice followed AAV delivery**. To score mtDNA-wide off-targeting in the mice, we analyzed mtDNA from hearts of vehicle-injected controls and mice injected with active or inactive versions of the DdCBE-Nd3-9577-1 pair. Vehicle-injected and catalytically inactive editor samples were used as a control in order to distinguish DdCBE-induced C:G-to-T:A single-nucleotide variants (SNVs) from natural background heteroplasmy. The average frequencies of mtDNA-wide off-target C:G-to-T:A editing by DdCBE-Nd3-9577-1 in the adult animals treated for 3 weeks were comparable to those of the vehicle-injected and catalytically inactive-DdCBE controls (0.026–0.046%) (Supplementary Fig. 3a). However, the adult mice treated with DdCBE-Nd3-9577-1 for 24 weeks showed ~7-fold higher average off-target editing frequency (0.22-0.30%) as compared with the controls (Supplementary Fig. 3a). We observed the highest off-target editing in the neonates treated with DdCBE-Nd3-9577-1, which was ~3-fold higher than the one observed for 24-week AAV-treated adult mice (Supplementary Fig. 3a). There was a positive correlation

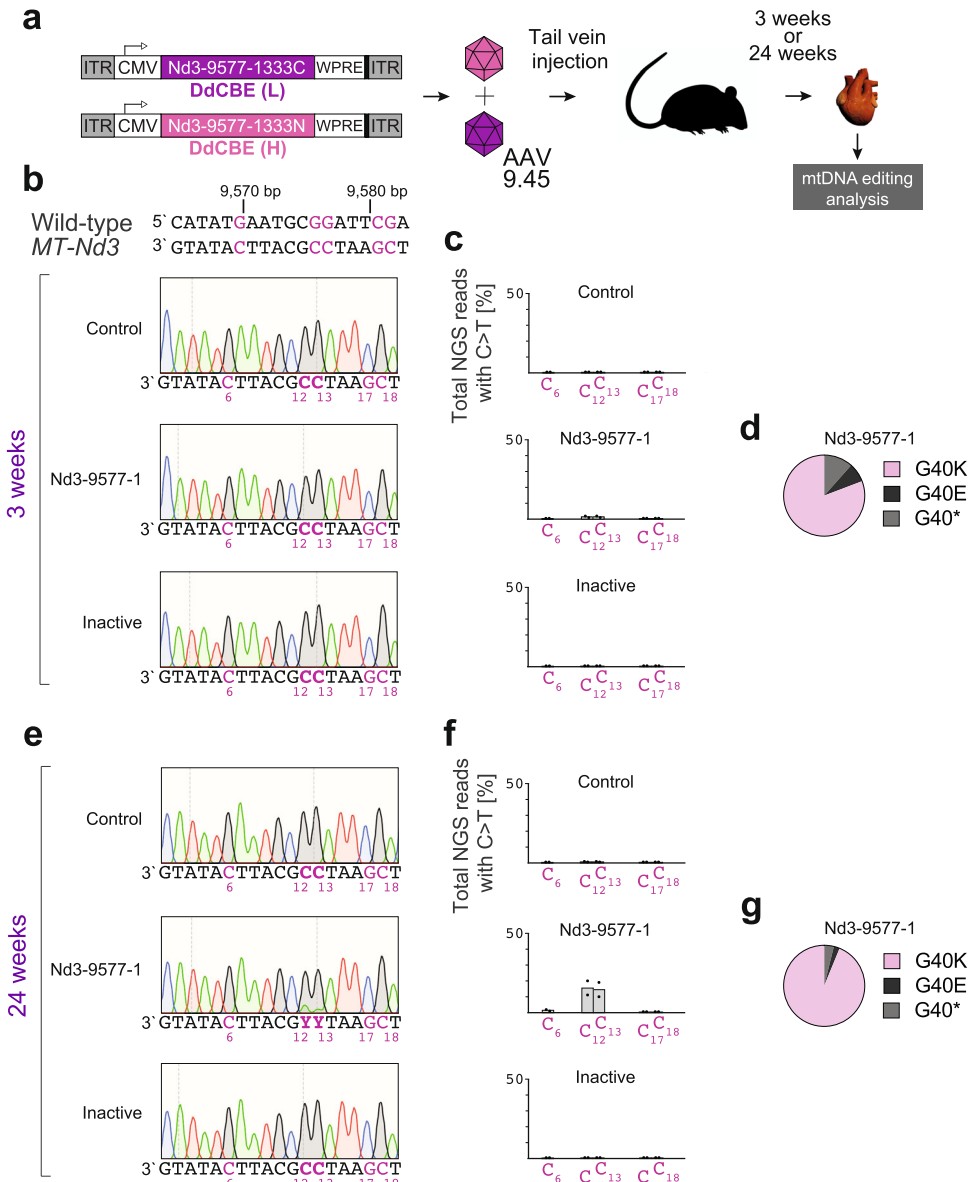

**Fig. 3 Mitochondrial DNA editing in adult mouse hearts. a** Scheme of in vivo experiments with adult mice. The DdCBE-Nd3-9577-1 monomers (see Fig. 2), and their catalytically inactive versions, were encoded in separate AAV genomes, encapsidated in AAV9.45 then simultaneously administered by tail-vein (TV) injection at $1 \times 10^{12}$ vg/mouse of each monomer. Animals were sacrificed either 3 or 24-weeks post-injection and their cardiac tissue was examined for mtDNA editing. **b, e** Editing of mouse *MT-Nd3* with DdCBE in mouse heart at 3-weeks (**b**) or 24-weeks (**e**) post-injection, analyzed by Sanger sequencing. Potential editing sites are indicated in purple. **c, f** The NGS analysis of the DdCBE editing within the targeted region in adult mouse hearts at 3 weeks (**c**) or 24 weeks (**f**) after injection. Bars represent the mean ($n = 2$). Source data are provided as a Source Data file. **d, g** The distribution of NGS reads containing m.9576 G ($C_{12}$) or m.9577 G ($C_{13}$) edits at 3 weeks (**d**) or 24 weeks (**g**) after injection. The G40K reads contain both m.9576 G > A ($C_{12} > T_{12}$) and m.9577 G > A ($C_{13} > T_{13}$) mutations, G40E reads contain only the m.9577 G > A ($C_{13} > T_{13}$) mutation, while G40* reads contain only the m.9576 G > A ($C_{12} > T_{12}$) mutation. Source data are provided as a Source Data file.

between on-target and off-target editing, with increased m.9576 G and m.9577 G modification being accompanied by higher levels of C:G-to-T:A SNVs (Supplementary Fig. 3b). Next, we tested off-target editing at nuclear pseudogenes, which are identical or share a high degree of sequence homology with mtDNA (nuclear mitochondrial DNAs, NUMTs). We did not observe editing beyond the background at the tested NUMTs, even though one of them was identical with the mtDNA on-target sites (Supplementary Fig. 4). This result is consistent with previous reports of exclusive mitochondrial localization of DdCBEs[28]. Taken together, our data show that, while no detectable off-targeting is observed in nDNA upon AAV delivery of DdCBE, substantial

off-targets are observed in mtDNA, especially when the on-target modification is also high. The latter result suggests that future development of DdCBE must focus on further improvement of the precision of this emerging tool.

## Discussion
The discovery of a bacterial cytidine deaminase acting on double-stranded DNA (DddA) led to the development of mitochondrial DddA-derived cytosine base editors (DdCBEs), which is likely to revolutionize the field of mammalian mtDNA genetic modification[28]. The DdCBE technology provides the potential to reverse engineer the mitochondrial genome in animal cells and

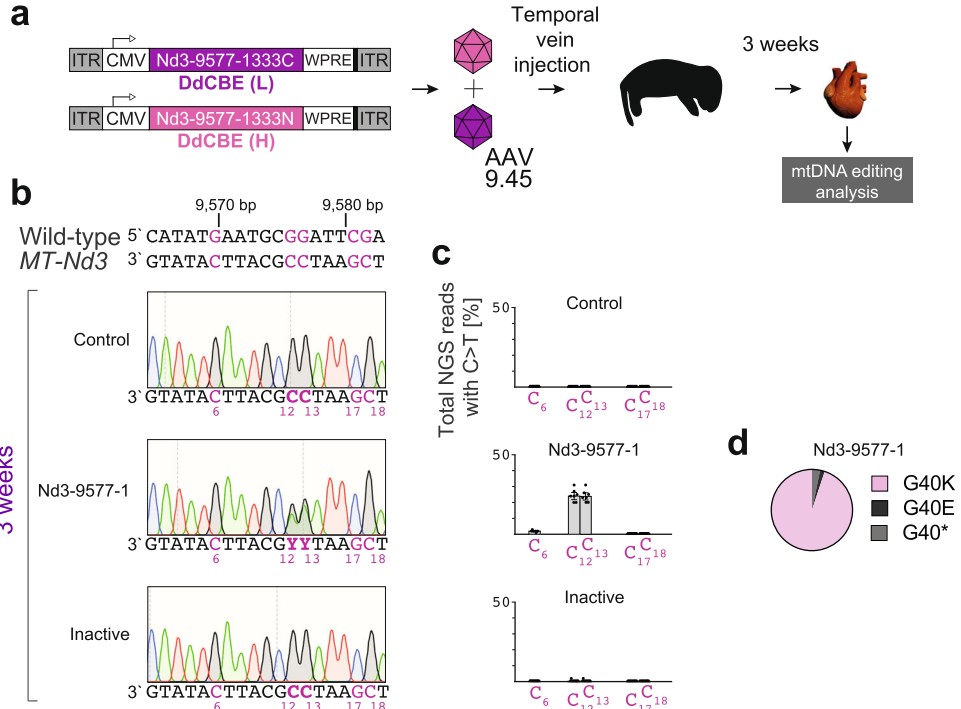

**Fig. 4 Mitochondrial DNA editing in neonatal mouse hearts.** **a** Scheme of in vivo experiments with neonatal mice. The DdCBE-Nd3-9577-1 monomers (see Fig. 2), and their catalytically inactive versions, were encoded in separate AAV genomes, encapsidated in AAV9.45 then simultaneously administered by temporal vein injection at $1 \times 10^{12}$ vg/mouse of each monomer. Animals were sacrificed 3-weeks post-injection and their cardiac tissue was examined for mtDNA editing. **b** Editing of mouse *MT-Nd3* with DdCBE in neonatal mouse heart at 3-weeks post-injection, analyzed by Sanger sequencing. Potential editing sites are indicated in purple. **c** The NGS analysis of the DdCBE editing within the targeted region in neonatal mouse hearts. Bars represent the mean and error bars represent ±SEM (n = 7). Source data are provided as a Source Data file. **d** The distribution of NGS reads containing m.9576 G ($C_{12}$) or m.9577 G ($C_{13}$) edits in neonatal hearts at 3-weeks post-injection. The G40K reads contain both m.9576 G > A ($C_{12} > T_{12}$) and m.9577 G > A ($C_{13} > T_{13}$) mutations, G40E reads contain only the m.9577 G > A ($C_{13} > T_{13}$) mutation, while G40* reads contain only the m.9576 G > A ($C_{12} > T_{12}$) mutation. Source data are provided as a Source Data file.

eventually correct homo- and heteroplasmic pathogenic point mutations in mtDNA. Generation of novel animal models is now expected to proceed in an expedited fashion, either by DdCBE-mediated manipulation of Embryonic Stem (ES) cells, direct modification of mouse embryos or, as shown here, by somatic delivery. As with any emerging technologies, the DdCBE approach needs to be tested in multiple systems to validate its usefulness. Thus far, the versatility of DdCBEs was shown by successfully base editing five mtDNA genes with efficiencies ranging between 5 and 50% in human cells in vitro[28]. Further proof-of-concept exemplified the use of base editing in mouse embryos and reported successful germline transmission of DdCBE-induced mtDNA edits[29]. In the latter report, mutations in *MT-Nd5*, generated by delivering DdCBE mRNAs in mouse zygotes were maintained throughout development and differentiation. These mutations were successfully transmitted to off-spring (F1) with heteroplasmy levels of up to 26%, providing evidence that DdCBEs can be used to generate mouse models with bespoke mtDNA mutations[29]. In this study, we used somatic AAV delivery of DdCBEs, providing a proof-of-concept for an alternative in vivo mtDNA mutagenesis means. The presented successful AAV-based mtDNA editing is also critical for in vivo proof-of-concept and insights into potential clinical translation to human somatic mitochondrial gene correction therapies to treat primary mitochondrial disease (PMD) phenotypes in worst-affected tissues. These future therapies could be crucial for mtDNA-associated PMDs, whose de novo genetics and hard-to-predict penetrance make preimplantation genetic diagnosis (PGD) screening difficult.

We predicted two possible issues that could prevent efficient mtDNA editing by DdCBEs in vivo. [i] The editing of mitochondrial genomes with the current DdCBEs involves inhibition of mitochondrial base excision repair (BER), to allow the retention of uracil in DNA (the result of cytosine deamination) and a sufficient level of active mtDNA replication allowing for the conversion of uracil into thymine[28]. The efficacy of mitochondrial BER had not been fully investigated in vivo and it was possible that it operates at the level of preventing efficient DdCBE-mediated editing. [ii] Previous reports highlighted that mammalian mtDNA is replicated continuously even in post-mitotic cells[33], but whether its activity is sufficient to achieve successful editing by DdCBEs was unknown. The successful mtDNA editing in mouse hearts reported here show that [i] the efficacy of BER operating in mouse heart does not lead to immediate excision of uracil in DNA, precluding efficient editing, and [ii] the level of mtDNA replication is high enough for fixing C-to-U deamination events into C-to-T mutations within 3 or 24 weeks after DdCBE treatment of neonatal or adult mice, respectively.

In neonates, our data also show that DdCBE administration earlier during development results in higher editing efficiency. A partial explanation for this finding could be that [i] at earlier vector administration, the greater AAV vector-to-cell ratio promoted higher transduction of the DdCBEs in the neonate heart and [ii] the postnatal development of mouse cardiomyocytes involves substantial mtDNA replication leading to an almost 13-fold increase of mtDNA copy number during the first four weeks of postnatal life[34], increasing the probability of fixing C-to-U deamination events into C-to-T mutations. Considering future

therapeutic interventions based on DdCBE-mediated mtDNA correction, this observation may be encouraging in the context of those mitochondrial diseases with early-onset and a rapid progression, for which administration of potential therapies in adults would be inadequate. Nonetheless, further studies on the gene correction potential of AAV-delivered DdCBEs would be required in mtDNA-disease mouse model. However, none of the four currently available mouse models harboring pathogenic mtDNA point mutations is suitable for a DdCBE-mediated gene correction study either due to the nature of mutation (*MT-ND6*: m.13997 G > A, *MT-MK*: m.7731 G > A, *MT-TA*: m.5024 C > T) or incompatible sequence context upstream of a T-to-C mutation (*MT-COI*: m.6589 T > C)[35]. The latter means that, in addition to DdCBE-based approaches, the field should continue generating novel mouse mtDNA mutated mouse lines using the established, phenotype-first pipeline, as it can generate any mtDNA mutations (not only C:G to T:A offered by DdCBEs)[36].

Our results confirm the previous observation that each DddA$_{tox}$ split edits the TC sites with a preference for specific windows in the spacing region. They also highlight that for any given target sequence, testing G1397 and G1333 splits in both orientations are required to achieve on-target editing. Here we demonstrate that the pair DdCBE-Nd3-9577-1 combining the G1333-C split in the TALE targeting the L-strand with G1333-N on the TALE targeting the H-strand produces more robust base editing of $C_{12}$ and $C_{13}$ located on the H-strand. Such observations fall in line with the previous reported preference of this split combination, which favors base editing of H-strand Cs present in the center of the editing window between the TALEs[28].

Our off-target editing analysis revealed that adult mice treated with DdCBE-Nd3-9577-1 for 24 weeks show ~0.25% of C:G-to-T:A SNV frequencies mtDNA-wide, which is higher than these reported previously for DdCBEs transiently expressed for 3 days in human cells, which ranged between ~0.05 to ~0.15%[28]. Also, the C:G-to-T:A SNV off-target frequencies detected in the neonates injected with the active DdCBE-Nd3-9577-1 pair for 3 weeks were on average at ~0.8%, which was ~5 times higher than the least "precise" mitochondrial base editor reported by Mok et al.[28]. We attribute these differences to longer mtDNA-DdCBE exposure time (weeks vs days) in our experiments and conclude that further optimization of mitochondrial DdCBE concentration and specificity will be required, especially in long-term in vivo experiments.

In the present study *MT-Nd3* G40K was installed to provide a proof-of-concept of somatic mtDNA editing. However, this mutation is located in the conserved ND3 loop involved in active/deactive state transition[31,32]. It is expected that high G40K heteroplasmy will result in mitochondrial dysfunction by permanently locking complex I in active confirmation, which warrants further studies. In addition, it has been previously shown that targeting the reversible S-nitrosation of the neighboring ND3 residue C39[37] protects against ischaemia-reperfusion (IR) injury. In this line, the G40K mutant could be explored in the context of changes in exposure of Cys39 in models of IR injury.

In conclusion, DdCBE is a promising tool for de novo mtDNA editing in post-mitotic tissue, which upon further research and optimization could be used to revert pathogenic mtDNA variants in patients affected with mitochondrial disease.

## Methods

**Ethics statement**. All animal experiments were approved by the local Animal Welfare Ethical Review Body (AWERB) at the University of Cambridge and carried out in accordance with the UK Animals (Scientific Procedures) Act 1986 (Procedure Project Licence: P6C20975A) and EU Directive 2010/63/EU.

**Plasmid construction and viral vectors**. The DdCBE architectures used were as reported in[28]. A catalytically dead DddA$_{tox}$ (E1347A) was used in the "inactive" control experiments. TALE arrays were designed using the Repeat Variable

Diresidues (RVDs) containing NI, NG, NN, and HD, recognizing A, T, G, and C, respectively. To construct the plasmids used in the cell screen, all DdCBEs ORFs were synthesized as gene blocks (GeneArt, Thermo Fisher) and cloned into pVax vectors downstream of a mitochondrial localization signal (MLS) derived from SOD2, using the 5′ KpnI and 3′ BglII restriction sites (Supplementary Sequences 1). Vector construction of DdCBEs intended for AAV production was achieved by PCR amplification of the transgenes to include 5′ NotI and 3′ BamHI sites, allowing cloning into a rAAV2-CMV backbone (Supplementary Sequences 2), previously reported in[26]. The resulting plasmids were used to generate recombinant AAV2/9.45 viral particles at the UNC Gene Therapy Center, Vector Core Facility (Chapel Hill, NC).

**Cell culture and transfections**. NIH/3T3 cells (CRL-1658$^{TM}$, American Type Culture Collection (ATCC)) were cultured at 37 °C under 5 % (vol/vol) CO$_2$ and in complete Dulbecco's Modified Eagle Medium (DMEM) (4.5 g/L glucose 2 mM glutamine, 110 mg/ml sodium pyruvate), supplemented with 10% calf bovine serum with iron and 5% penicillin/streptomycin (all from Gibco). Mycoplasma tests in the culture medium were negative. The cell line was not authenticated in this study. For DdCBE pair screen, NIH/3T3 mouse cells plated in six-well tissue culture plates at a confluency of 70% were transfected with 3200 ng of each monomer (L and H), to a total of 6400 ng of plasmid DNA using 16 µl of FuGENE-HD (Promega), following manufacturer′s guidelines. After 24 h, cells were collected for Fluorescence-activated cell sorting (FACS) and sorted for GFP and RFP double-positive cells using a BD FACSMelody$^{TM}$ Cell sorter. The collected double-positive cells were allowed to recover for another 6 days and then used for DNA extraction, as described below.

**Animals**. Mice in a C57BL/6J background were obtained from Charles River Laboratories. The animals were maintained in a temperature- and humidity-controlled animal care facility with a 12 h light/12 h dark cycle and free access to water and food, and they were sacrificed by cervical dislocation. In adult experiments, 8-week-old male mice were administered systemically by tail-vein injection with $1 \times 10^{12}$ AAV particles of each monomer [AAV- DdCBE (L) - Nd3-9577 - G1333-C and AAV- DdCBE (H) - Nd3-9577- G1333-N]. An equal dose was applied in newborn pups (Postnatal day 1—males and females) via the temporal vein, using a 30 G, 30° bevelled needle syringe. Control mice were injected with similar volumes of vehicle buffer (1× PBS, 230 mM NaCl and 5% w/v D-sorbitol).

**Genomic DNA isolation and Sanger sequencing of *MT-Nd3* locus**. NIH/3T3 mouse cells were collected by trypsinization, washed once in PBS, and re-suspended in lysis buffer (1 mM EDTA, 1% Tween 20, 50 mM Tris (pH = 8)) with 200 µg/ml of proteinase K. Lysates were incubated at 56 °C with agitation (300 RPM) for 1 h, and then incubated 95 °C for 10 min before use in downstream applications. Genomic DNA from mouse heart samples (~50 mg) was extracted with a Maxwell® 16 Tissue DNA Purification Kit in a Maxwell® 16 Instrument (Promega), according to the manufacturer′s instructions.

For Sanger sequencing, the *MT-Nd3* edited region was PCR-amplified with GoTaq G2 DNA polymerase (Promega) using the following primers: Mmu_Nd3_Fw: 5′- GCA TTC TGA CTC CCC CAA AT -3′; and Mmu_Nd3_Rv: 5′- GGC CTA GAG ATA GAA TTG TGA CTA GAA -3′. The PCR was performed with an initial heating step of 1 min at 95 °C followed by 35 cycles of amplification (30 s at 95 °C, 30 s at 63 °C, 30 s at 72 °C), and a final step of 5 min at 72 °C. PCR purification and Sanger sequencing were carried out by Source Bioscience (UK) with the Mmu_Nd3_Rv primer.

**High-throughput targeted amplicon sequencing**. Genomic DNA was extracted as described above. For high-throughput targeted amplicon resequencing of the *MT-Nd3* region, a 15,781 bp fragment was first amplified by long-range PCR to avoid amplification of nuclear mtDNA pseudogenes (NUMTs), with PrimeSTAR GXL DNA polymerase (TAKARA) using the following primers: Long-R_mtDNA_Fw: 5′- GAG GTG ATG TTT TTG GTA AAC AGG CGG GGT -3′; and LongR_mtDNA_Fw: 5′-GGT TCG TTT GTT CAA CGA TTA AAG TCC TAC GTG -3′. The PCR was performed with an initial heating step of 1 min at 94 °C followed by 10 cycles of amplification (10 s at 98 °C, 13 min at 68 °C), and a final step of 10 min at 72 °C. All PCR reactions from this and the following steps were cleaned up AMPure XP beads (Beckman Coulter, A63881). An aliquot of the purified long-range PCR reactions was amplified with primers containing an overhang adapter sequence, compatible with Illumina index and sequencing primers, in a 25 µl reaction, with Phusion® Hot Start Flex 2X Master Mix (NEB, M0536S) using the following primers: Nd3adapter_Fw: 5′– TCG TCG GCA GCG TCA GAT GTG TAT AAG AGA CAG TTC TGA ATA AAC CCA GAA GAG AGT -3′; and Nd3adapter_Rv: 5′- GTC TCG TGG GCT CGG AGA TGT GTA TAA GAG ACA GGG TAG ACG TGC AGA GCT TGT -3′. The PCR was performed with an initial heating step of 30 s at 98 °C followed by 12 cycles of amplification (10 s at 98 °C, 15 s at 62.2 °C, 15 s at 72 °C), and a final step of 5 min at 72 °C. The indexing PCR was performed using the Nextera XT Index Kit (Illumina, FC-131-1096) according to the manufacturer′s instructions. The PCR was performed with an initial heating step of 30 s at 98 °C followed by 6 cycles of

amplification (30 s at 98 °C, 30 s at 55 °C, 30 s at 72 °C), and a final step of 5 min at 72 °C. Libraries were subjected to high-throughput sequencing using the Illumina MiSeq platform (PE75) and demultiplexed using the Illumina MiSeq manufacturer´s software.

For mtDNA-wide off-targets analysis, two overlapping long amplicons (8331 bp and 8605 bp) covering the full mtDNA molecule were amplified by long-range with PrimeSTAR GXL DNA polymerase (TAKARA) using the following primers: mmu_ND2_Fw: 5´- TCT CCG TGC TAC CTA AAC ACC -3´; with mmu_ND5_Rv: 5´- GGC TGA GGT GAG GAT AAG CA -3´; and mmu_ND2_Rv: 5´- GTA CGA TGG CCA GGA GGA TA -3´; with mmu_ND5_Fw: 5´- CTT CCC ACT GTA CAC CAC CA -3´. The PCR was performed with an initial heating step of 1 min at 94 °C followed by 16 cycles of amplification (30 s at 98 °C, 30 s at 60 °C, 9 min at 72 °C), and a final step of 5 min at 72 °C.

For nuclear DNA off-targets assessment, two regions with high homology to the MT-Nd3 targeted region were analyzed. A region with 100% identity in chromosome 1 was amplified by long-range with PrimeSTAR GXL DNA polymerase (TAKARA) using the following primers: G40K_NUMT100_Fw2: 5´- T GC ACT GCT GAC CCA TTA AT -3´; with G40K_NUMT100_Rv2: 5´- ACA CAC ACT AGA CAA CAC CCA -3´. The second region with 86% identity in chromosome 14 was amplified using the following primers: G40K_NUMT86_Fw1: 5´- CTG GTG GTC ACT TGG TGT GT -3´; with G40K_NUMT86_Rv1: 5´- TGT TAC ATG TTT CTC TGT TTT TGC T -3´.

The PCR was performed with an initial heating step of 1 min at 94 °C followed by 20 cycles of amplification (30 s at 98 °C, 30 s at 60 °C, 9 min at 72 °C), and a final step of 5 min at 72 °C. Tagmentation and the indexing PCR were performed using the Nextera XT Index Kit (Illumina, FC-131-1096) according to the manufacturer´s instructions and as described above. Libraries were subjected to high-throughput sequencing using the Illumina MiSeq platform (PE250) and demultiplexed using the Illumina MiSeq manufacturer´s software.

**Processing and mapping of high-throughput data.** Quality trimming and 3' end adaptor clipping of sequenced reads were performed simultaneously, using Trim Galore! (--paired)[38]. For targeted amplicon resequencing of the MT-Nd3 region and for mtDNA-wide off-targets analysis reads were aligned to ChrM of the mouse reference genome (GRCm38) with Bowtie2 (--very-sensitive; --no-mixed; --no-discordant)[39]. For nuclear DNA off-targets assessment reads were aligned to the two regions with high homology (GRCm38). Count tables for targeted amplicon resequencing of the MT-Nd3 region were generated with samtools mpileup (-q 30)[40] and varscan[41]. To study the editing pattern per reading and to determine the percentage of reads that had edits at $C_{12}$ and $C_{13}$, we used cutadapt (-e = 0; --action=none; --discard-untrimmed)[42] for the 10nt surrounding the editing site in combination with the samtools flagstat command. For the mtDNA-wide and nuclear DNA off-targets analysis REDItools2.0[43] was used (-bq 30) to generate count tables. The average mtDNA-wide C•G-to-T•A off-target editing frequency was assessed by summation of all off-target C-to-T and G-to-A editing frequencies divided by the total number of C•G sites in mouse mtDNA (5990).

**Quantification of viral genomes copy number and relative mtDNA content by quantitative real-time PCR.** All quantitative real-time PCR reactions were performed in a QuantStudio™ 3 system (Thermo Fisher), using a TaqMan™ Gene Expression Master Mix (Thermo Fisher), according to manufacturer´s instructions and Ct values obtained using the built-in software. Reactions were performed in a final volume of 20 µl in technical triplicates and each dataset represents samples from an individual mouse. Determination of viral genomes copy number of AAV-DdCBE (L) was performed using primers and probe detecting the DddA_tox split G1333-C: VG_1333C_Fw: 5´- GGA ACT TGC GGA TTT TGT GT -3´; VG_1333C_Rv: 5´- TTT GGG AGA ATT GGA GTT GC-3´; VG_1333C_probe: 5´-/56-FAM/ CGT AAA ACG /ZEN/ GGG AGC TAC AG /3IABkFQ/ -3´. Copy number of AAV-DdCBE (H) was performed using primers and probe detecting the DddA_tox split G1333-N: VG_1333N_Fw: 5´- CAT ACG CAC TTG GCC CTT AC -3´; VG_1333N_Rv: 5´- AAC ACC TTG CTC TCC AGT CC -3´; VG_1333N_probe: 5´-/56-FAM/ CAA CTG CCT /ZEN/ GCA TAC AAT GG /3IABkFQ/ -3´. DNA input was normalized to the nuclear gene Actin, using the primers and probe: Mmu_Actin_Fw: 5´- CTG CTC TTT CCC AGA CGA GG -3´; Mmu_Actin_Rv: 5´- AAG GCC ACT TAT CAC CAG CC -3´; Mmu_Actin_probe: 5´-/56-TAMN/ ATT GCC TTT CTG ACT AGG TG /3BHQ/ -3´. Standard curves were generated using tenfold serial dilutions of an individual plasmid DNA encoding both G1333-C and G1333-N splits, from $2.5 \times 10^{-6}$ to $10^{-3}$ ng/reaction. Copy numbers were obtained by plotting the logarithm of the plasmid quantity against the measured Ct values from samples.

For relative mtDNA content in heart samples, mtDNA levels were quantified using primers and probe detecting MT-ND1: Mmu_Nd1_Fw: 5´- GAG CCT CAA ACT CCA AAT ACT CAC T -3´; Mmu_Nd1_Rv: 5´- GAA CTG ATA AAA GGA TAA TAG CTA TGG TTA CTT CA -3´; Mmu_Nd1_probe: 5´-/56-FAM/ CCG TAG CCC /ZEN/ AAA CAA T /3IABkFQ / -3´. Levels of MT-ND1 were normalized to the nuclear gene Actin, using the primers and probe: Mmu_Actin_Fw: 5´- CTG CTC TTT CCC AGA CGA GG -3´; Mmu_Actin_Rv: 5´- AAG GCC ACT TAT CAC CAG CC -3´; Mmu_Actin_probe: 5´-/56-TAMN/ ATT GCC TTT CTG ACT AGG TG /3BHQ/ -3´. The Ct values for each sample were used to calculate relative mtDNA copy number using the ΔΔCt analysis, using vehicle-injected mice as control.

**Immunoblotting of DdCBEs in mouse hearts.** Heart samples (~50 mg) were homogenized in 200 uL of ice-cold RIPA buffer (150 mM NaCl, 1.0% NP-40, 0.5% sodium deoxycholate, 0.1% SDS, 50 mM Tris, pH 8.0) containing 1X cOmpleteTM mini EDTA-free Protease Inhibitor Cocktail (Roche, UK), using a gentleMACS™ dissociator. Homogenates were incubated on ice for 20 mins and then clear by centrifugation (20,000 × g for 20 min at 4 °C). Protein lysates (~20 µg) were mixed with 10 X NuPAGE™ sample reducing agent and 4X NuPAGE™ LDS sample buffer (Invitrogen), and incubated for 5 min at 95 °C. The boiled protein samples were then separated in a Bolt 4–12% Bis-Tris (Thermo Fisher) pre-cast gel and later transferred to a PVDF membrane using an iBlot 2 gel transfer system (Thermo Fisher), according to the manufacturer´s recommendations. The residual proteins that remained in the gel were detected using SimpleBlue SafeStain (Thermo Fisher) and used as a loading control. The membrane was blocked in 5% milk in PBS with 0.1% Tween 20 (PBS-T) for 1 h at room temperature (RT) and then incubated with either rat anti-HA-tag antibody (Roche, 11867423001), 1:1000 diluted in 5% milk in PBST or mouse anti-FLAG-M2-tag (Sigma–Aldrich, F3165), 1:2000 diluted in 5% milk in PBST. Membranes were washed three times with PBS-T for 10 min at RT and then incubated with HRP-linked secondary antibodies, either anti-rat IgG (Cell Signaling, 7077 S) or anti-mouse IgG (Promega, W4021), 1:5000 diluted in 5% milk in PBST. The membranes were washed another three times as before and imaged digitally with an Amersham Imager 680 blot and gel imager (GE Healthcare), upon incubation with Amersham ECL™ Western Blotting Detection Reagents (GE Healthcare).

**Immunohistochemistry and microscopy.** At sacrifice, mouse heart samples were frozen by immersion in isopentane cooled in liquid nitrogen. For immunohistochemistry analysis, heart samples were mounted in optimal cutting temperature compound (OCT) and sectioned on a cryostat at −20 °C to a thickness of 8 µm. Sections were fixed in 10% neutral buffered formalin (SIGMA, HT501128) at RT. After three washes in PBS, samples were permeabilized using 0.2% Triton X- 100 in PBS for 15 min at RT, followed by three washes in PBS, and then blocked in 5% normal goat serum + 2% BSA in PBS for 1 h at RT. After three washes in PBS, samples were incubated with the rabbit anti-HA-Tag (Cell Signaling, 3724) primary antibody 1:2000 in DAKO solution (Agilent, S3022) for 1 h at RT, followed by three additional washes in PBS. Sections were then incubated with goat anti-rabbit IgG (H + L) AlexaFluor diluted 1:300 in DAKO diluent solution 568 for 1 h at RT, washes three times in PBS and finally coverslip with prolonged diamond antifade mountant with DAPI (Thermo Fisher, P36962). All confocal images were acquired using a Zeiss LSM880 microscope using identical acquisition parameters.

**Statistics.** Graphical visualization of data and all statistical analyses were performed with GraphPad Prism software (version 8.0). All numerical data are expressed as mean ± standard error of mean (SEM). Ordinary one-way ANOVA with Dunnett's test was used for multiple comparisons. Animals were randomized and no blinding to the operator was used.

**Reporting summary.** Further information on research design is available in the Nature Research Reporting Summary linked to this article.

## Data availability
The authors declare that the data supporting the findings of this study are available within the paper and its supplementary information files, apart from proprietary scripts which are available upon request by contacting the authors. The NGS files generated in this study have been deposited in GEO: GSE184064. Source data are provided with this paper.

## Code availability
On and off-target effects from next-generation sequencing data were calculated with Varscan2, source code: https://github.com/Jeltje/varscan2, written by Koboldt et al.[41].

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

## Acknowledgements

We would like to acknowledge the members of the Mitochondrial Genetics Group (MRC-MBU, University of Cambridge) for useful discussion during the course of this research. We would like to thank Ahmed-Noor Agip for his help with Fig. 1c. This work was supported by core funding from Medical Research Council UK (MC_UU_00015/4). P.S.-P. and P.A.N. are additionally supported by The Champ Foundation and The Lily Foundation, respectively.

## Author contributions

P.S.-P. and M.M. planned and designed experiments; P.A.N. and C.D.M. performed the immunohistochemistry and NGS experiments; L.V.H. and P.S.-P. analyzed the NGS experiments; K.T. managed the animal work and performed the histology experiments; P.S.-P. performed the remaining experiments. P.S.-P. and M.M. drafted the manuscript. M.M. supervised the study. All authors revised the manuscript.

## Competing interests

M.M. is a co-founder, shareholder and member of the Scientific Advisory Board of Pretzel Therapeutics, Inc. P.S.-P., P.A.N., and C.D.M. provide consultancy services for Pretzel Therapeutics, Inc. The remaining authors declare no competing interests.
