## [Peer Review File · Nature Communications]

Reviewers' Comments:

Reviewer #1:

Remarks to the Author:

The mitochondrial DNA is required for the production of the main energy converter in the cells, the OXPHOS system. Placed within the mitochondrial matrix, this DNA has been notoriously difficult to manipulate. These difficulties to manipulate mitochondrial DNA in turn limits the curing of mutations in mitochondrial DNA that can cause severe human diseases as well as impairs detailed analyses into the mechanisms of mitochondrial gene expression. A previous study reported that double-strand specific deminases can be targeted to specific sites within mitochondrial DNA to evoke base-editing on specific codons (Nature 583, 631-637). In this work, Silva-Pinheiro et al. now provide in a short, concise manuscript proof of concept that this strategy can be employed for targeted base editing also in living animals. Specifically, they directed a split deaminase to a specific site in the ND3 gene. Transfection into cultured cells or transfection with AAV vectors into mice enabled efficient editing in both models.

Overall, the manuscript is very well written and reports an important proof-of-principle experiment that will substantially change research into mammalian mitochondrial gene expression. All experiments are performed at high standards including the appropriate controls. I would suggest acceptance as it is.

Reviewer #2:

Remarks to the Author:

In this manuscript, the authors applied the newly developed mitochondrial base editors, namely DdCBEs, to introduce desired mutations into the mitochondrial genome in adult and neonatal mice after delivery via adeno-associated virus. Although this proof-of-concept study is an exciting development for the genome editing field, I feel the significance of this paper, at least in its current form, is not high enough for publication in Nature Communications.

Major comments:

1. The authors focus on the ND3 m.9576G and m.9577G loci, which are not necessarily clinically relevant. The authors do mention possible applications of the introduced mutations, but without solid supporting evidence. Also, as the authors mention, previous studies using different mtDNA editors (mtZFNs, mitoTALENs) have achieved rescue of disease phenotypes. Therefore, the authors should focus on targeting pathological sites to increase the impact of the study.

2. Related to comment 1, if the authors were to show phenotypical improvement using mitochondrial gene editing in an animal model of a mitochondrial disease, the study would be exciting enough for publication in Nat. Commun.

3. It would be interesting to know the genome-wide off-target effects of TALE DdCBEs and the level of byproduct editing in nuclear DNA, which should be examined in detail.

Minor comments:

1. In the supplementary figures, the immunohistochemistry images need additional staining markers for mitochondria, such as MitoTracker.

2. The labeling of graphs in Figure 2~4 may be a bit confusing for readers. I feel that Figures 2c, 2d, 3c, 3d, 3f, 3g, 4c, and 4d should also have labels (DdCBE-Nd3-9577-1~4 or control/DdCBE-Nd3-9577/Inactive) on the corresponding graphs.

3. Many sentences need citations. Below are some examples.

"Mitochondria play a central role in energy provision to the cell and in several key metabolic pathways, such as thermogenesis, calcium handling, iron-sulphur cluster biogenesis and apoptosis."

"Mitochondrial diseases are genetic disorders, caused by mutations, either in nDNA or mtDNA, that lead to mitochondrial energy production impairment and perturbations in other aspects of cellular homeostasis."

"Mammalian cells can contain 100s to 1000s copies of mtDNA. Pathogenic variants in mtDNA can either present in all copies (homoplasmy) or only in a proportion of genomes (heteroplasmy), with

mutant load varying across cells, tissues, and organs.”

“However, recently a novel tool has emerged: DddA-derived cytosine base editor (DdCBE), which catalyzes site-specific C:G to T:A conversions in mtDNA with good target specificity in human cultured cells.”

“Such observations fall in line with the previous reported preference of this split combination, which favors base editing of H-strand Cs present in the center of the editing window between the TALEs.”

5. Figure 1d is a part of Figure 1c; I think these two parts should be combined as a single figure subsection.

6. Some abbreviations, such as ES cell, RVDs, RT, etc., are not explained or are inconsistently used. Additionally, abbreviations for tail vein (TV) and temporal vein (TeV) are not necessary. “Tail vein” should be used instead of “TV” in Figure 3a, and “temporal vein” should be used instead of “TeV” in Figure 4a.

Reviewer #1 (Remarks to the Author):

The mitochondrial DNA is required for the production of the main energy converter in the cells, the OXPHOS system. Placed within the mitochondrial matrix, this DNA has been notoriously difficult to manipulate. These difficulties to manipulate mitochondrial DNA in turn limits the curing of mutations in mitochondrial DNA that can cause severe human diseases as well as impairs detailed analyses into the mechanisms of mitochondrial gene expression. A previous study reported that double-strand specific deaminases can be targeted to specific sites within mitochondrial DNA to evoke base-editing on specific codons (Nature 583, 631-637). In this work, Silva-Pinheiro et al. now provide in a short, concise manuscript proof of concept that this strategy can be employed for targeted base editing also in living animals. Specifically, they directed a split deaminase to a specific site in the ND3 gene. Transfection into cultured cells or transfection with AAV vectors into mice enabled efficient editing in both models.

Overall, the manuscript is very well written and reports an important proof-of-principle experiment that will substantially change research into mammalian mitochondrial gene expression. All experiments are performed at high standards including the appropriate controls. I would suggest acceptance as it is.

Thank you very much for this positive assessment of our study and for recommending our manuscript for publication.

Reviewer #2 (Remarks to the Author):

In this manuscript, the authors applied the newly developed mitochondrial base editors, namely DdCBEs, to introduce desired mutations into the mitochondrial genome in adult and neonatal mice after delivery via adeno-associated virus. Although this proof-of-concept study is an exciting development for the genome editing field, I feel the significance of this paper, at least in its current form, is not high enough for publication in *Nature Communications*.

We are grateful for the Reviewer for finding our *proof-of-concept study exciting*. Below and in the revised version of the manuscript we are providing additional data requested by Reviewer and further explanations of challenges we face in the mitochondrial field in terms of suitable *in vivo* models.

Major comments:

1. The authors focus on the ND3 m.9576G and m.9577G loci, which are not necessarily clinically relevant. The authors do mention possible applications of the introduced mutations, but without solid supporting evidence.

We agree with the Reviewer that the m.9576G and m.9577G loci are not clinically relevant and, in our manuscript, we have not claimed that they were. We used these sites to show, for the first time, that DdCBEs can operate in post-mitotic tissues *in vivo*, which in our view is an important finding that merits publication in *Nature Communications*.

DdCBEs are expected to be an important tool to manipulate mtDNA in mammals. However, as every emerging technology, DdCBEs needs to be tested in different models. For this reason, we repurposed existing TALE domains from our collection, for which we had unpublished evidence of specific binding to mouse mtDNA at positions: m.9549-9564 and m.9584-9599. That way, in case of negative results for the application of AAV-encapsidated DdCBEs, we could reduce the possibility that the lack of measurable DdCBE editing is due to TALEs not binding to the targets. In this context, the nature of the mutations introduced was merely a technical consideration, given the binding abilities of the available TALE array.

However, as we already note in the manuscript, despite ND3 p.G40K being installed as a proof-of-concept, it is located in the conserved ND3 loop involved in active/deactive state transition. It is expected that high p.G40K heteroplasmy will result in mitochondrial dysfunction by permanently locking complex I in an active confirmation, which can be studied by others interested in the topic. In addition, it has been shown previously the targeting the reversible S-nitrosation to the ND3 Cys39 residue next to the modified residue led to protection against ischaemia-reperfusion (IR) injury. Here

again, the p.G40K mutant could be useful to study changes in exposure of Cys39 in models of IR injury. We mentioned this possible future use in our manuscript; however, we feel that these experiments will be beyond the scope of our study.

Also, as the authors mention, previous studies using different mtDNA editors (mtZFNs, mitoTALENs) have achieved rescue of disease phenotypes. Therefore, the authors should focus on targeting pathological sites to increase the impact of the study.

2. Related to comment 1, if the authors were to show phenotypical improvement using mitochondrial gene editing in an animal model of a mitochondrial disease, the study would be exciting enough for publication in Nat. Commun.

We agree with the Reviewer that it is important that future base editing experiments should aim to correct pathogenic mtDNA and rescue disease phenotypes, thus paving a way towards their therapeutic use. However, unfortunately, at present there is no mouse model that could be corrected using the current base editing technology.

The historical inability to modify mtDNA sequences in mammalian mitochondria has hindered the development of *in vivo* models. Only four (!) mouse models for mtDNA point mutations have been reported thus far. These “transmitochondrial” mouse models have been generated by the introduction of mitochondria containing naturally occurring variants in their mtDNA into embryonic stem (ES) cells, which were subsequently introduced into early embryos to create chimeric animals. Another method involved female germline transmission of mtDNA mutations introduced by the proofreading-deficient PolGA in combination with a multi-step breeding program, limiting the number of transmitted mutations, to select mouse lines with clonally expanded heteroplasmic point mutations based on screening based on OXPHOS function.

These four models contain the following homo- or heteroplasmic mutations:

MT-COI m.6589T>C - Kasahara *et al.* Hum Mol Genet 15, 871-881 (2006).

MT-ND6 m.13997G>A - Lin *et al.* Proc Natl Acad Sci U S A 109, 20065-20070 (2012).

MT-MK m.7731G>A - Shimizu A *et al.* Proc Natl Acad Sci U S A. 111(8):3104-3109 (2014).

MT-TA m.5024C>T - Kauppila *et al.* Cell Rep 16, 2980-2990, (2016).

While m.6589T>C could, in principle, be corrected through deamination of cytosine at the mutation site, the current DdCBEs can only deaminate cytidines in the TC sequence context, which m.6589T>C is lacking. The *MT-ND6*, *MT-MK* and *MT-TA* G:C to A:T mutations cannot be targeted by cytoside base editors.

Therefore, none of the existing mouse model of mitochondrial disease could be used to achieve phenotypic improvement using mitochondrial gene editing with DdCBEs. Nonetheless, our experiments, despite being provided in WT animals, provide *in vivo* proof-of-concept and insights into potential clinical translation to somatic mitochondrial gene correction therapies to treat mtDNA disease phenotypes once suitable models are developed.

To address the Reviewer’s comments, we discuss future pre-clinical experiments involving DdCBEs in the context of the current lack of suitable *in vivo* models in the revised manuscript.

3. It would be interesting to know the genome-wide off-target effects of TALE DdCBEs and the level of byproduct editing in nuclear DNA, which should be examined in detail.

Following Reviewer’s suggestion, we performed off-target assessment in mtDNA and nDNA – Supplementary Figure 3 and 4, respectively.

To score mtDNA-wide off-targeting, we followed the NGS protocols published in the initial DdCBE paper by Mok *et al.* (Mok *et al.* Nature 2020). DNA from wild-type mouse hearts and hearts from mice injected with inactive version of the DdCBE-Nd3-9577-1 pair were used as a control in order to distinguish DdCBE-induced C:G-to-T:A single-nucleotide variants (SNVs) from background heteroplasmy. While, in the adult animals exposed with AAV DdCBE-Nd3-9577-1 for 3 weeks, the off-

target editing was undetectable when compared to the control samples. In contrast, the adults treated with DdCBE-Nd3-9577-1 for 24 weeks showed ~0.25% of C:G-to-T:A SNV frequencies, which was higher than these reported previously for DdCBEs transiently expressed (3 days) in human cells, which ranged between ~0.05 to ~0.15% (Mok *et al.* Nature 2020). However, the C:G-to-T:A SNV off-target frequencies detected in the neonates injected with the active DdCBE-Nd3-9577-1 pair for 3 weeks were on average at ~0.82%, which was ~5 times higher than the least “precise” mitochondrial base editor reported for Mok *et al.* 2020.

We also assessed editing two nDNA sites, which were identical (chromosome 1) or closely resembled (chromosome 14) to the targeted mtDNA locus. We did not detect any off-targeting of these nDNA loci, consistent with DdCBEs being efficiently imported to mitochondrial of the cardiac tissue after AAV delivery.

We included the off-target analysis data in the revised manuscript and changed the text to discuss the higher levels off-targeting in the AAV *in vivo* experiments as compared to *in vitro* editing. We also discuss potential reasons for this result, hypothesising that it is due to the long exposure time and sub-optimal concentration of AAV DdCBEs. We discuss the general requirement for extensive optimisation of DdCBE designs to limit off-targeting.

Minor comments:

1. In the supplementary figures, the immunohistochemistry images need additional staining markers for mitochondria, such as MitoTracker.

The main aim of experiments presented in Suppl. Figure 1 and 2 was to show that DdCBEs are expressed upon AAV delivery. We appreciate the reviewer’s comment that it could be beneficial to evaluate DdCBEs’ localisation *in vivo*. We note that mitochondrial import of the current DdCBE architecture was optimised and studied in cultured cells (Mok, *et al* 2020). Furthermore, MitoTracker has been designed to label mitochondria in live cells when added to culture media. In this work, we stained snap frozen heart samples that have been fixed for histochemistry analyses, therefore, MitoTracker is not adequate. In addition, cardiac tissue is densely packed with mitochondria resulting in widespread signal across the section. Therefore, there is no resolution to properly assess mitochondrial co-localisation using this method. For the same reason, the use of antibody-based staining of mitochondria (for example with anti-TOM20), also results in labelling of the entire cardiac tissue. If the Reviewer’s concern is the possible localisation of DdCBEs in the nucleus resulting in nuclear DNA off-targets, we addressed this point above and concluded that there is no detectable nuclear DNA off-targets.

2. The labeling of graphs in Figure 2~4 may be a bit confusing for readers. I feel that Figures 2c, 2d, 3c, 3d, 3f, 3g, 4c, and 4d should also have labels (DdCBE-Nd3-9577-1~4 or control/DdCBE-Nd3-9577/Inactive) on the corresponding graphs.

Following Reviewer’s recommendation, we have now provided the labels.

3. Many sentences need citations. Below are some examples.

Following Reviewer’s comment, we provided suitable references.

“Mitochondria play a central role in energy provision to the cell and in several key metabolic pathways, such as thermogenesis, calcium handling, iron-sulphur cluster biogenesis and apoptosis.” Mitochondrial disorders as windows into an ancient organelle. Vafai SB, Mootha VK. Nature. 491:374-83. (2012)

Mitochondrial diseases: the contribution of organelle stress responses to pathology
A Suomalainen, BJ Battersby Nature Reviews Molecular Cell biology 19 (2), 77-92 (2018)

“Mitochondrial diseases are genetic disorders, caused by mutations, either in nDNA or mtDNA, that lead to mitochondrial energy production impairment and perturbations in other aspects of cellular homeostasis.”

Gorman, G. S. et al. Mitochondrial diseases. Nat. Rev. Dis. Primers 2, 16080 (2016).

“Mammalian cells can contain 100s to 1000s copies of mtDNA. Pathogenic variants in mtDNA can either present in all copies (homoplasmy) or only in a proportion of genomes (heteroplasmy), with mutant load varying across cells, tissues, and organs.”

Filigrana, R., Mennuni, M., Alsina, D. & Larsson, N. G. Mitochondrial DNA copy number in human disease: the more the better? *FEBS Lett.* 595, 976–1002 (2021).

Stewart, J. B. & Chinnery, P. F. Extreme heterogeneity of human mitochondrial DNA from organelles to populations. *Nat. Rev. Genet.* 22, 106–118 (2021).

“However, recently a novel tool has emerged: DddA-derived cytosine base editor (DdCBE), which catalyzes site-specific C:G to T:A conversions in mtDNA with good target specificity in human cultured cells.”

Mok, B. Y., de Moraes, M. H., Zeng, J., Bosch, D. E., Kotrys, A. V., Raguram, A., ... & Liu, D. R. A bacterial cytidine deaminase toxin enables CRISPR-free mitochondrial base editing. *Nature*, 583(7817), 631-637 (2020).

“Such observations fall in line with the previous reported preference of this split combination, which favors base editing of H-strand Cs present in the center of the editing window between the TALEs.”

Mok, B. Y., de Moraes, M. H., Zeng, J., Bosch, D. E., Kotrys, A. V., Raguram, A., ... & Liu, D. R. A bacterial cytidine deaminase toxin enables CRISPR-free mitochondrial base editing. *Nature*, 583(7817), 631-637 (2020).

5. Figure 1d is a part of Figure 1c; I think these two parts should be combined as a single figure subsection.

This has been corrected.

6. Some abbreviations, such as ES cell, RVDs, RT, etc., are not explained or are inconsistently used. Additionally, abbreviations for tail vein (TV) and temporal vein (TeV) are not necessary. “Tail vein” should be used instead of “TV” in Figure 3a, and “temporal vein” should be used instead of “TeV” in Figure 4a.

This has been corrected.

Reviewers' Comments:

Reviewer #2:

Remarks to the Author:

The authors addressed all of my concerns. I think that the paper is now suitable for publication.